# Storage Location Assignment for Improving Human–Robot Collaborative Order-Picking Efficiency in Robotic Mobile Fulfillment Systems

**Yue Chen ***  **and Yisong Li**

School of Economics and Management, Beijing Jiaotong University, Beijing 100044, China; ysli@bjtu.edu.cn
* Correspondence: cyuelinda@foxmail.com; Tel.: +86-010-51687175

**Abstract:** The robotic mobile fulfillment (RMF) system is a parts-to-picker warehousing system and a sustainable technology used in human–robot collaborative order picking. Storage location assignment (SLA) tactically benefits order-picking efficiency. Most studies focus on the retrieval efficiency of robots to solve SLA problems. To further consider the crucial role played by human pickers in RMF systems, especially in the context that the sustainable performance of human workers should be paid attention to in human–robot collaboration, we solve the SLA problem by aiming to improve human–robot collaborative order-picking efficiency. This study specifically makes decisions on assigning multiple items of various products to the slots of pods in the RMF system, in which human behavioral factors are taken into account. To obtain the solution in one mathematical model, we propose the heuristic algorithm under a two-stage optimization method. The results show that assigning correlated products to pods improves the retrieval efficiency of robots compared to class-based assignment. We also find that assigning items of each product to slots of pods, considering behavioral factors, benefits the operation efficiency of human pickers compared to random assignment. Improving human–robot collaborative order-picking efficiency and increasing the capacity usage of pods benefits sustainable warehousing management.

**Keywords:** human–robot collaboration; sustainable technology; parts-to-picker warehousing system; behavioral factors; robotic mobile fulfillment (RMF) system; storage location assignment; order picking

## 1. Introduction

Order picking is one of the most laborious activities in e-commerce warehouses [1–3]. To mitigate the overwhelming workloads of human pickers, human–robot collaborative order picking is increasingly used in warehouses [4,5]. The robotic mobile fulfillment (RMF) system is a representative of human–robot collaborative order-picking systems, which enables human pickers to collaborate with mobile robots in order picking [6], as shown in Figure 1. In RMF systems, mobile robots drive the pods from the storage area to the workstations and human pickers subsequently pick the required items from the pods [7], as shown in Figure 2a. RMF systems are supported by sustainable technologies in Industry 4.0, including automated guided navigation, artificial intelligence, and autonomous robots [8,9]. These sustainable technologies applied in RMF systems stimulate the process of human–robot collaborative order picking.

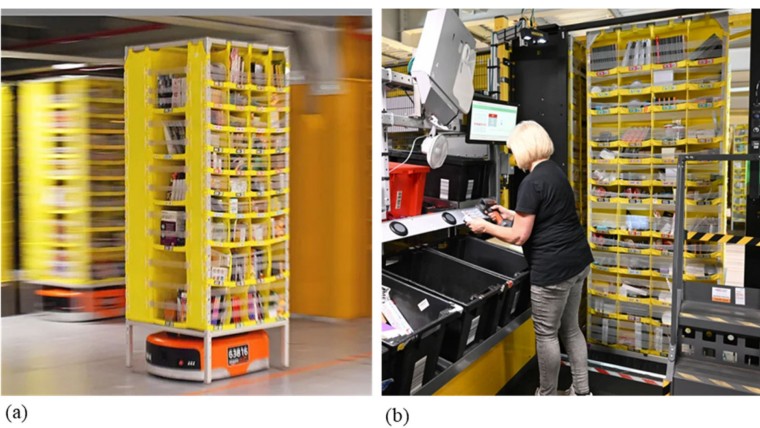

**Figure 1.** (**a**) A pod driven by a mobile robot in RMF system. (**b**) Human–robot collaborative order picking in RMF system [10,11].

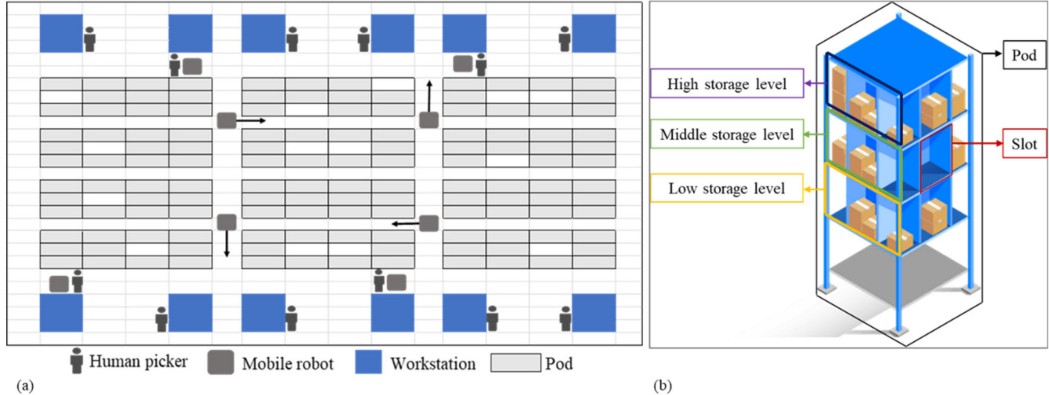

**Figure 2.** (**a**) The layout of an RMF system (**b**) A pod in RMF system.

Compared to manual order picking, the automation in human–robot collaborative order picking largely improves operation efficiency, which benefits the sustainable energy consumption in RMF systems. On the other hand, the flexibility of human pickers in human–robot collaboration further enhances sustainable efficiency. A lot of studies focus on the operation efficiency of robots in human–robot collaboration [12]. However, the role of human pickers in human–robot collaboration draws less attention. From the perspective of sustainable operations in RMF systems, the labor-effectiveness of human pickers is as important as the energy-efficiency of robots. For the sustainable improvement of order-picking efficiency in RMF systems, we pay more attention to operation efficiency of human pickers in human–robot collaboration.

Storage location assignment (SLA) plays an important role in improving order-picking efficiency in RMF systems [13]. The SLA problem is commonly defined as assigning products to storage locations [14]. In RMF systems, storage locations consist of pods and slots. Pods are located in the storage area and multiple slots are placed at different storage levels of a pod, as shown in Figure 2b. Assigning storage to pods in RMF systems has been widely researched in the SLA problem [15–19]. Correlated storage assignment is widely adopted for reducing the number of retrievals of pods [20]. With correlated storage assignment, products that are frequently ordered together are assigned to the same pod. Due to various products being commonly requested in an order, multiple pods are needed to be retrieved for picking an order. Products which are stored in one pod can also be picked together in one retrieval.

However, assigning storage to the slots draws little attention in current research. Slots are placed at different storage levels of a pod, which results in various behaviors in order picking. Human pickers need to bend to pick items from low storage levels (Figure 3a) or

stretch and climb to pick items from high storage levels (Figure 3c). Picking items from the middle storage level (Figure 3b) is relatively effortless because it is at a similar height to human pickers. These various behaviors result in the operation efficiency of human pickers varying to different degrees. Existing studies neglect that storage assignment in slots has an impact on human pickers' operation efficiency [21]. We therefore define the SLA problem in this study as assigning storage to the slots of pods in RMF systems for the purpose of improving the order-picking efficiency of human–robot collaboration.

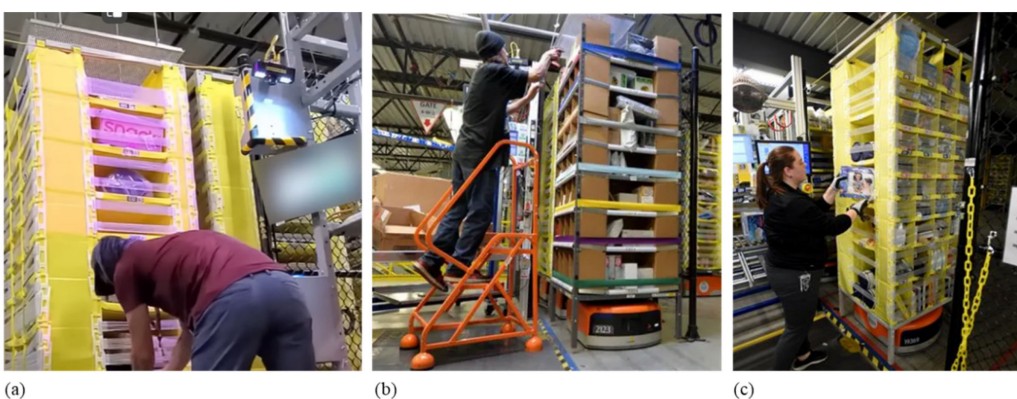

**Figure 3.** (**a**) Bending to pick. (**b**) Stretching to pick. (**c**) Reaching to pick [22–24].

As storage assignment in slots placed on different storage levels leads to different human behaviors in order picking, we consider the behavioral factors of human pickers to solve the SLA problem proposed in this study. The behavioral factors of order picking can be categorized as location factors and product factors [25]. Location factors are also defined as 'golden zone picking', which is the storage area between a picker's waist and shoulders [26]. The 'golden zone' is most beneficial for order-picking performance because it reduces pickers' fatigue. Taking an instance in Figure 2b, the middle storage level is the 'golden zone' for human pickers. Given that slots placed on different storage levels of pods lead to different order-picking behaviors, we consider storage levels as the location factor in RMF systems. In addition, the weight and volume of storage are commonly regarded as product factors which have an effect on the physical effort of human pickers [27]. For example, pickers spend more physical effort on grabbing heavy and big products than grabbing light and small products. Products stored in e-commerce warehouses are of various weights and volumes, so the operation efficiency of picking those products is different. We therefore consider the product factors of weight and volume in the storage assignment.

With the common definition of the SLA problem that is assigning storage to pods in RMF systems, we define the SLA problem in this study as storage assignments to the slots of pods. Given that multiple items are commonly stored for each product in e-commerce warehouse, assigning items of products is considered in the SLA problem. Furthermore, we propose the following research questions: (1) how to assign products to pods, considering the correlated relationship between products, and (2) how to assign items of each product to slots which are placed on different storage levels of pods, considering human behavioral factors in order picking. For the purpose of improving the retrieval efficiency of pods driven by robots and the operational efficiency of human pickers, we built a mathematical model to solve this SLA problem. To answer these research questions in one model increases the complexity of obtaining the solution. We consequently propose a two-stage optimization method to solve the model, with each stage to answer each research question, in which correlated storage assignment and strategies of balancing grabbing effort are adopted to design the heuristic algorithms. This study contributes to solving the SLA problem from the perspective of assigning items of products to the slots of pods, in which behavioral factors in order picking are taken into account. Most studies solve the SLA problem to improve the retrieval efficiency of pods. This study obtains the solution of the SLA problem,

aiming to improve order-picking efficiency in human–robot collaboration, including the retrieval efficiency of pods and the operation efficiency of human pickers.

The remainder of this paper is organized as follows: In Section 2, studies related to the SLA problem are reviewed. Section 3 describes the SLA problem and formulates it into a mixed-integer mathematical model. The solution approach of the SLA problem is designed in Section 4. We discuss the numerical results in Section 5. Conclusions and future research are presented in Section 6.

## 2. Literature Review

This paper studies the SLA problem of RMF systems for improving the retrieval efficiency of pods and the operation efficiency of human pickers. In this section, we first review the literature on sustainable warehousing and human–robot collaborative order picking in RMF systems and generally analyze human behavioral factors in order picking. Subsequently, we narrow it down to the storage location assignment, specifically focusing on correlated storage assignment and class-based storage assignment. A few studies on considering human behavioral factors in storage assignment are further summarized.

### 2.1. Sustainable Warehousing and Human–Robot Collaborative Order Picking in RMF Systems

Warehouses are one of the principal contributors to the rise in greenhouse gas emissions. Sustainable warehousing accordingly draws a lot of attention in recent research. The implementation of Industry 4.0 in warehouses accelerates the process to achieve the Sustainable Development Goals (SDGs), including SDG 7 (Affordable and Clean energy), SDG 9 (Industry, Innovation, and Infrastructure), and SDG 12 (Ensure Sustainable Consumption and Production Patterns), proposed by the United Nations. Beltrami et al. [28] summarize Industry 4.0 technologies applied for sustainability practices, in which human–robot collaboration technology is increasingly used for sustainable production, sustainable performance measurement, and sustainable management. Aravindaraj et al. [29] conducted a content analysis in a literature review, and research which ways warehouse management might benefit from the deployment of Industry 4.0 under the SDGs. In their findings, hundreds of reviewed studies suggest that the improvement of order-picking efficiency, driven by Industry 4.0 technology, benefits sustainable warehousing management. Bartolini et al. [30] systematically reviewed the literature on sustainable warehousing, in which storage assignment for automatic warehousing systems is one of the classifications. In our context, we specifically focus on solving the problem of storage location assignment in RMF systems which deploy multiple Industry 4.0 technologies for the purpose of improving human–robot collaborative order-picking efficiency. This also benefits sustainable warehousing in the long term.

In the field of storage assignment in RMF systems, numerous studies focus on improving order-picking efficiency driven by robots, such as decreasing travel distance, reducing travel time, and decreasing the number of retrievals. However, the factors of human pickers in human–robot collaboration draw relatively less attention in this field. A few studies quantify the dynamic physical working rate of human pickers into the optimization model in the context of order picking [31]. Wang et al. [32] formulate dynamic human pickers' working rate to design a work schedule in human–robot coordinated order picking. The result shows that the schedule that takes the fluctuations in a picker's working rate into account reduces the expected total picking time. Shew et al. [33] propose a human–robot coordinated order-picking process in the consideration of human pickers' fatigue. They demonstrate that the proposed scheme can alleviate pickers' fatigue without much influence on order-picking efficiency. The fluctuated working efficiency of human pickers are considered in existing studies. However, how the behavioral factors of human pickers play a role in order-picking efficiency is neglected. To particularly consider the role of human pickers in sustainable warehousing, behavioral factors in human–robot collaboration are taken into account in this study.

### 2.2. Human Behavioral Factors in Order Picking

Various studies investigate the factors of human pickers in order-picking operations. Battini et al. [34] consider human energy expenditure, which is affected by item characteristics, item popularity, order profiles, and physical dimensions of shelves, and integrate the energy expenditure rate into time estimation in order-picking optimization. Elbert et al. [35] research the effect of route deviation resulting from behavioral factors and evaluate order-picking efficiency, taking human pickers' behavior into account. In these studies, the physical aspects of human pickers have a major impact on the estimation of order-picking efficiency, since order-picking tasks consist of continuous physical activities. The outcome of completing the required physical activities could be evaluated based on the order cycle time, which is one of the key indicators to evaluate the efficiency of operational systems [36].

In RMF systems, the repetitive physical activities at pick stations involve grabbing items from the pods and putting the picked items into the package. The order-picking time also reveals the physical efforts spent on these physical activities. Greater physical effort leads to a longer order-picking time. To quantify the physical efforts spent by human pickers, the operational time for conducting physical activities at pick stations is used to measure the operational efficiency of human pickers in this study.

To include the physical aspects of human pickers in storage assignment with respect to the order-picking time as the outcome, only a few studies that analyze how the physical aspects of human pickers affect the order-picking time can be referred to. A framework to incorporate human factors in order-picking planning models is built up by Grosse et al. [37,38]. In this framework, they mention that the physical aspects of human pickers in pick activities involve stretching, bending, reaching for items, extracting, grabbing, picking, and putting down items. Larco et al. [25] conclude and classify the key factors involved in these physical activities as location factors and product factors. In this study, picking levels are included into the location factors, and the quantity picked, the unit mass of the product, and the unit volume of the product are included into the product factors. The storage assignment model is subsequently constructed with the bi-objective of minimizing workers' discomfort caused by the investigated factors and minimizing the order-picking time. In the context of order picking, one location factor is also defined as 'golden zone picking'. Petersen et al. [26] consider the layout-dependent pick time in storage assignment to take advantage of golden-zone picking, in which high-demand items should be stored in the area between a picker's waist and shoulders to improve picking performance and to reduce picker fatigue.

To further overview the location factors and product factors that affect the order-picking time, we also refer to the literature in the field of material handling in assembly line. Finnsgård et al. [27] observe that factors including the part size and weight and the vertical pick distance impact the speed of manual picking significantly in assembly lines. Several empirical research studies on picking in assembly lines mention that the pick time depends on the weight of the items [39,40]. One location factor is also described as using two hands to grab at the height of the shoulders or knees in handling operations by St-Vincent et al. [41].

In this study, we consider the storage level as a location factor and the weight and volume as product factors in storage location assignment in RMF systems.

### 2.3. Storage Location Assignment

Storage location assignment concerns assigning products to storage locations and has a major impact on order-picking efficiency [10]. Class-based storage assignment is commonly used in practice and has been widely researched in existing studies. In class-based storage assignment, products are classified according to the turnover rate and are assigned to a storage area based on their classification. The class of storage locations with high turnover rate is located near to the depot and products are assigned randomly within each class [42]. Accorsi et al. [43] assigns products with high demand to available storage locations in an attempt to reduce the travel distance for optimizing order-picking efficiency. Manzini et al. [44] model class-based storage assignment over life cycle picking patterns

and determine the storage capacity for storage classes, considering the assignment of SKUs to the storage classes. Ang et al. [45] sequentially group storage locations into a number of classes and determine the number of arriving pallets of products that are assigned to the class in a certain period. Cezik et al. [46] classify units according to the velocity and divide the velocity classes for the pods in the storage area. They found that a three-class policy achieves more travel-time reduction than a two-class policy. Yuan et al. [47] show a similar result: that class-based storage with two or three classes can achieve most of the potential benefits. Yu et al. [48] prove that any small number of classes is near-optimal in class-based storage assignment, and ABC classification is effective in practice.

Some research on correlated storage assignment regards assigning products with a high correlated relationship to the same storage location. Mirzaei et al. [49] adopt correlated dispersed storage assignment in RMF systems and suggest that it can improve picking performance compared with random and turnover-based storage policies. With the definition of correlation in correlated storage assignment, products that are frequently ordered together have high degrees of correlation. The concepts of affinity, similarity, and association are also used to describe the correlated relationship between products, the same as correlation. To assign products to a storage location according to their correlation, the degree of correlation between products is commonly calculated based on historical orders. Pang et al. [50] present a data mining-based algorithm for storage location assignment, in which the association relationships between different products are extracted from customer orders. The results show that the proposed algorithm is efficient at decreasing total travel distances compared with closest-open-location allocation and dedicated storage allocation. Zhang et al. [51] apply the FP-Growth algorithm to describe demand correlation patterns among items to solve the storage assignment allocation problem. Glock et al. [52] use two indices, the correlation of order demand and the pick frequency in order picking, to calculate correlation between products. Most research involves constructing mathematical models for correlated storage assignment, with the decision variable of storage location assignment and the objective of maximizing the correlation. The methodologies to obtain such a solution include assigning pairs of products, assigning clusters of products, and swapping the assigned storage locations. Flamand et al. [53] consider affinities between certain pairs of assorted categories to assign products to retail shelves for the purpose of stimulating customers to purchase combined products that are stored near to each other. Xiang et al. [54] built a storage assignment model to decide which product to put in which pod for maximizing the similarity between products stored in the same location. Kim et al. [55] consider item reassignment in RMF systems in order to maximize the sum of similarity values of items in each pod. Li et al. [56] calculate the degree of association between items which are combined in a correlated cluster, and the cluster is assigned to a pod.

To summarize the literature on class-based storage assignment and correlated storage assignment, correlated storage assignment concerns the storage location for pairs of products and class-based storage assignment focuses on a single product's assignment based on the turnover rate. Due to the fact that multiple products are commonly required in an order in e-commerce warehouses, we adopt correlated storage assignment in RMF systems. Existing studies regard the correlation degree between pairs of products as the indicator in the storage assignment decision process. We additionally calculate the aggregated correlation degree of products that have been assigned to the same pod to make decisions on storage assignment.

In summary, with respect to the physical aspects of human pickers, location factors and product factors are largely investigated in existing research but are rarely applied to order-picking optimization. Larco et al. [25] is the only study both investigating these factors and including these factors in storage assignment. In this study, we specifically focus on the storage assignment in RMF systems. This study incorporates location factors and product factors in the process of assigning storage to the slots of pods. Referring to the concept of golden-zone picking, we define storage levels which locate slots with different reaching heights within pods as the location factor. We also consider the weight and volume of products as the product factors.

## 3. Problem Description and Formulation

In this section, we describe the SLA problem proposed in this study and formulate it in a mathematical model.

### 3.1. Problem Description

The SLA problem is generally described as assigning products to storage locations. In RMF systems, storage assignments to slots which are placed on different storage levels results in different operation efficiencies of human pickers. For example, picking from the slots on the middle storage levels costs less time than picking from high and low storage levels. We therefore define the SLA problem in RMF systems as storage assignments to the slots of pods.

The purpose of solving the SLA problem is to improve both the retrieval efficiency of pods and the operation efficiency of human pickers. Most studies solve the SLA problem only for decreasing the retrieval time of pods driven by robots. However, the time spent by human pickers at workstations is neglected. The retrieval time of pods for picking an order depends on the storage assignment to pods, in which products that are stored in a pod can be picked together in one retrieval. The grabbing time spent by human pickers depends on which slot to pick from. In this study, we include both the time of grabbing products and the time of retrieving pods into the objective of SLA optimization.

To estimate the grabbing time spent by human pickers, we assume that more grabbing effort commonly results in a longer grabbing time. We therefore measure the grabbing time based on the degrees of effort spent on physical activities. To introduce grabbing effort in RMF systems, the physical activities of human pickers at a workstation can be decomposed into three steps: reaching the storage level to pick from; searching the slot placed on the storage level and using hands to grab the product; and putting the picked product in the order basket. The storge level to pick from affects how much effort human pickers spend. To explain this in detail, we take the instances of low, middle, and high storage levels, as shown in Figure 2b. Human pickers need to bend to pick from low storage levels and stretch or even climb up a ladder to pick from high storage levels. The middle storage level is at a similar height to human pickers, so human pickers can easily access and spend the least effort to pick from there. As such, the storage levels at different heights in pods results in various degrees of grabbing effort spent by human pickers in order picking. Additionally, the product to pick also affects the grabbing effort spent by human pickers. More specifically, the weight and volume of a product impact the grabbing effort. For example, grabbing heavy products costs more energy for human pickers than grabbing light products; human pickers might manage to grab small products using one hand, whereas they use two hands to grab big products. Therefore, products with different units of weight and volume also lead variation in the grabbing effort of human pickers. To summarize, the storage levels and the weight and volume of products affect the grabbing effort of human pickers in RMF systems.

However, multiple items are commonly stored for the same product in warehouses since the number of items that is required for the same product varies from order to order. Therefore, we consider that multiple items of the same product can be stored in the same pod. With the assumption that human pickers grab one item of a product at a time, we obtain the grabbing time for order picking by aggregating the grabbing time for all required items in an order. To calculate the time for grabbing items with various weights and volumes from low/middle/high storage levels, we set a base line for the grabbing time. The base grabbing time is defined as the time for picking an item with one unit weight and one unit volume at the middle storage level (at a similar height to human pickers) of pods. As such, the grabbing time for any item (with any weight and volume and stored at any storage level of pods) can be calculated according to the base grabbing time in the formulation, rather than setting the grabbing time for each item as the parameter.

To evaluate the retrieval efficiency of pods, we measure the retrieval time of pods with the constant speed of mobile robots. Therefore, the retrieval time is related to the number

of retrievals of pods. Customers commonly request more than one product in an order and multiple pods are retrieved for meeting the requirements of each order. If multiple products commonly ordered together can be stored in one pod, those products can be picked in one retrieval, so the retrieval efficiency of pods can be improved in RMF systems. Products that are commonly requested together in an order are defined as correlated products. The higher correlation between products that are frequently ordered together can be found from historical orders. In correlated storage assignment, products are assigned to storage pods according to the correlation between products. With correlated storage assignment, the retrieval efficiency can be improved due to the decrease in the retrievals of pods. Hence, assigning correlated products to a pod is adopted to solve the SLA problem in this study to improve the retrieval efficiency of pods.

To improve the operation efficiency of human pickers, we balance the grabbing effort spent on each storage level to assign items to slots. Picking items from high or low storage levels costs more grabbing time than picking items from the middle storage level, but on the other hand, picking light and small items costs less time than picking heavy and big items. If light and small items are stored on the low/high storage levels and heavy and big items are stored on the middle storage level, the grabbing effort spent on picking all items from the low, middle, and high storage levels of a pod can be balanced. Therefore, the total grabbing time of picking all items from a pod can decrease when taking the balanced grabbing effort spent on each storage level into account. As such, assigning items of products to the storage levels of pods with a balanced grabbing effort plays an important role in improving grabbing efficiency.

To summarize, we resolve the SLA problem by making decisions on which products are stored together in the same pod and on which slots to store the items of each product. The SLA problem is commonly regarded as the tactical decisions in a warehouse and is operationally verified in order picking. Therefore, we jointly make order-picking decisions on which pods to retrieve for each order pick and which slots to reach for when grabbing the required items in an order.

### 3.2. Mathematical Model

We construct a mathematical model to solve the SLA problem defined in this study. The objective is to minimize the order-picking time (including the retrieval time of pods and the grabbing time of items). The decisions, including assigning products to pods, assigning items of each product to slots, retrieving pods to workstations for order picking, and picking items from the slots, are made in the mathematical models. As the storage levels on which the slots are placed make a difference on human pickers' operation efficiency, we make the decision of assigning items of each product to each storage level. This is regarded the same as the assignment to slots but is less complex to solve.

We make the following assumptions:

1.  Multiple items of a product are assigned to the same pod;
2.  The storage locations of pods are fixed in the storage area;
3.  The pod is retrieved to the workstation over the shortest distance. The retrieval distance is calculated by Manhattan distance;
4.  Multiple products can be picked in one retrieval and human pickers grab one item of a product at a time;
5.  The pods in a warehouse are the same and are all available at the beginning of storage assignment;
6.  The same number of slots are placed on each storage level and the slots are the same size;
7.  Items of the same product have the same weight and volume and are assigned to slots placed on the same storage level.

The parameters and decision variables involved in the mathematical model are shown in Table 1.

**Table 1.** Description of the parameters and decision variables.

| Parameters | Description |
|---|---|
| $J$ | The set of products. $J = \{1, 2, 3, \cdots\}$ |
| $I_j$ | The set of items of product $j \in J$. $I_j = \{1, 2, 3, \cdots\}$ |
| $P$ | The set of pods. $P = \{1, 2, 3, \cdots\}$ |
| $L$ | The set of storage levels. $L = \{1, 2, 3, \cdots\}$ |
| $O$ | The set of orders. $O = \{1, 2, 3, \cdots\}$ |
| $w_i$ | The units of weight of item $i \in I_j$, $j \in J$. $w_i = 1, 2, 3, \cdots$ |
| $v_i$ | The units of volume of item $i \in I_j$, $j \in J$. $v_i = 1, 2, 3, \cdots$ |
| $\alpha$ | The influence coefficient of item's weight on grabbing time. $\alpha \geq 0$ |
| $\beta$ | The influence coefficient of item's volume on grabbing time. $\beta \geq 0$ |
| $\gamma$ | The influence coefficient of storage level on grabbing time. $\gamma \geq 0$. |
| $t_p$ | The retrieval time of pod $p \in P$ to the closest workstation. $t_p \geq 0$ |
| $t_{base}$ | The base grabbing time of the item with one unit weight and one unit volume stored on the slot placed on the middle storage level (easiest to access). $t_{base} = 1, 2, 3, \cdots$ |
| $d_p$ | The travel distance between pod $p \in P$ in the storage area and the closet workstation. $d_p \geq 0$ |
| $v$ | The driving speed of mobile robots. $v \geq 0$ |
| $D_{j,o}$ | The number of required items of product $j \in J$ in order $o \in O$. $D_{j,o} = 1, 2, 3, \cdots$ |
| $s_j$ | The number of items of product $j \in J$ to be allocated on pods. $s_j = 1, 2, 3, \cdots$ |
| $M$ | The maximum number of products stored on a pod. $M = 1, 2, 3, \cdots$ |
| $N$ | The maximum number of items stored on a pod. $N = 1, 2, 3, \cdots$ |
| $W$ | The maximum units of weight allowed on a storage level. $W = 1, 2, 3, \cdots$ |
| $V$ | The maximum units of volume allowed on a storage level. $N = 1, 2, 3, \cdots$ |
| $C_{j,j'}$ | The correlation coefficient between product $j \in J$ and product $j' \in J$, $j \neq j'$. |
| $C_{j,p}$ | The correlation coefficient between product $j \in J$ to be assigned and the products that have been assigned to pod $p \in P$ |

| Decision Variables | Description |
|---|---|
| $x_{j,p}$ | $x_{j,p} \in \{0, 1\}$: If product $j \in J$ is assigned to the pod $p \in P$, $x_{j,p} = 1$. Otherwise, $x_{j,p} = 0$. |
| $y_{i,j,l}$ | $y_{i,j,l} \in \{0, 1\}$: If item $i \in I_j$ of product $j \in J$ is assigned to storage level $l \in L$ of a pod, $y_{i,j,l} = 1$. Otherwise, $y_{i,j,l} = 0$. |
| $z_{p,o}$ | $z_{p,o} \in \{0, 1\}$: If pod $p \in P$ is retrieved to pick the required products in order $o \in O$, $z_{p,o} = 1$. Otherwise, $z_{p,o} = 0$. |
| $q_{i,j,l}$ | $q_{i,j,l} \in \{0, 1\}$: If item $i \in I_j$ of product $j \in J$ on storage level $l \in L$ is picked, $q_{i,j,l} = 1$. Otherwise, $q_{i,j,l} = 0$. |

The objective function and constraints of the model are as follows:

$$Min \sum_{o \in O} \sum_{p \in P} x_{j,p} \cdot z_{p,o} \cdot t_p + \sum_{j \in J} \sum_{i \in I_j} y_{i,j,l} \cdot q_{i,j,l} \cdot (\alpha \cdot w_i + \beta \cdot v_i + \gamma \cdot l) \cdot t_{base} \quad (1)$$

$$\sum_{j \in J} x_{j,p} \leq M, \ p \in P \quad (2)$$

$$\sum_{j \in J} x_{j,p} \cdot s_j \leq N, \ p \in P \quad (3)$$

$$\sum_{i \in I_j} \sum_{j \in J} w_i \cdot y_{i,j,l} \leq W, \ l \in L \quad (4)$$

$$\sum_{i \in I_j} \sum_{j \in J} v_i \cdot y_{i,j,l} \leq V, \ l \in L \quad (5)$$

$$y_{i,j,l} \leq x_{j,p}, \ i \in I_j, \ j \in J, \ p \in P \quad (6)$$

$$z_{p,o} \leq \sum_{i \in I_j} \sum_{j \in J} \sum_{l \in L} x_{j,p} \cdot y_{i,j,l}, \; p \in P, \; o \in O \tag{7}$$

$$\sum_{i \in I_j} z_{p,o} \cdot y_{i,j,l} \cdot q_{i,j,l} = D_{j,o}, \; j \in J, \; p \in P, \; o \in O \tag{8}$$

$$\sum_{p \in P} \sum_{l \in L} x_{j,p} \cdot y_{i,j,l} = 1, \; i \in I_j, \; j \in J \tag{9}$$

$$C_{j,p} = \sum_{j' \in J} x_{j',p} \cdot C_{j,j'}, \; j \in J, \; j' \in J, \; j \neq j', \; p \in P \tag{10}$$

$$C_{j,p'} \leq x_{j,p} \cdot C_{j,p}, \; j \in J, \; p \in P, \; \forall p' \in P, \; p \neq p' \tag{11}$$

$$t_p = \frac{d_p}{v}, \; p \in P \tag{12}$$

$$x_{j,p} \in \{0,1\} \tag{13}$$

$$y_{i,j,l} \in \{0,1\} \tag{14}$$

$$z_{p,o} \in \{0,1\} \tag{15}$$

$$q_{i,j,p} \in \{0,1\} \tag{16}$$

Objective (1) is minimizing the total order-picking time, including the retrieval time of pods and the grabbing time of items of products. The grabbing time of an item is calculated according to the base line, $t_{base}$, and depends on the item's weight, volume, and the storage level to pick from. Constraint (2) indicates that, at most, $M$ products can be stored on a pod. Constraint (3) states that, at most, $N$ items can be stored on a pod. Constraint (4) requires that the total weight of items stored on a storage level cannot be more than $W$ units. Constraint (5) limits that the total volume of items stored on a storage level cannot be more than $V$ units. Constraint (6) ensures that once an item is assigned to a storage level, the decision of the assignment to the pod has been made. Constraint (7) stipulates that once a pod is retrieved to the workstation for meeting the requirements of order $o$, the number of items stored for the required product is at least one. Constraint (8) makes sure that the number of items picked from all retrieved pods meets the request of order $o$. Constraint (9) guarantees that each item of each product is assigned to a storage level of a pod. Equation (10) indicates that the correlation between product $j \in J$ (to be assigned) and the products that are assigned to pod $p \in P$ is the aggregated correlations between pairs of products ($j$ and $j'$). Constraint (11) indicates that if product $j \in J$ is assigned to pod $p \in P$, the correlation to pod $p \in P$ is higher than any other pods. Equation (12) is the calculation of the retrieval time of pods. The other constraints, (13)–(16), define the binary variables.

## 4. Solution Approach

The SLA problem is defined as assigning items of products to slots of pods in this study. To decrease the complexity of solving this problem, we formulate it as assigning items of products to the storage levels of pods in the mathematical model. However, the SLA problem is still an NP-hard problem, since we consider assigning the correlated products to the same pod by comparing the correlation between pairs of products. Additionally, the items to be assigned are of various weights and volumes, which increases the complexity of assignment within the limits of weight and volume on a storage level. To solve the NP-hard SLA problem described in this paper, we propose a two-stage optimization method for storage assignment.

### 4.1. Solution Framework Design

Due to the NP-hardness of the SLA problem, the decisions on assigning products to pods and assigning items to storage levels cannot be solved simultaneously. Pods are retrieved, and subsequently, human pickers pick items from storage levels in the order-picking process. We therefore propose a two-stage optimization method which corresponds to these two procedures in the order-picking process. The first stage of optimization is making decisions on assigning products to pods, in which correlated storage assignment is

adopted. Based on the decisions of the first stage of optimization, assigning items of each product to storage levels is decided in the second stage of optimization. The two-stage method proposed in this study enables feasible solutions to be obtained in a limited time. The strategies used in the heuristic algorithm design of the two-stage method mean that the solutions can be compared, and management insights can be derived from these solutions. The heuristic algorithms designed in the two-stage optimization method are coded in Python 3.11 and run on an X64-PC with Intel i5-1240P and a 1.70 GHz CPU.

Figure 4 shows the solution framework of storage assignment and order picking. In the first stage of optimization, correlation between products is calculated based on historical orders, in which the FP-Growth algorithm is adopted. The FP-Growth algorithm is advantageous for processing large datasets with many transactions and items [57]. As it fits into the features of e-commerce order sets, the FP-Growth algorithm can be efficiently used to calculate correlation. The second stage of optimization is based on greedy heuristics [58]. With greedy heuristics, the storage assignment of each item can be locally solved until the solutions of all items' assignments are derived. The optimization of the second stage aims to decrease the total grabbing time of all order picking. We propose strategies for balancing the grabbing time at each storage level to design the heuristic algorithms. The solutions of storage assignment are the input of order picking, which decides pod retrieval and item picking according to the requests in an order.

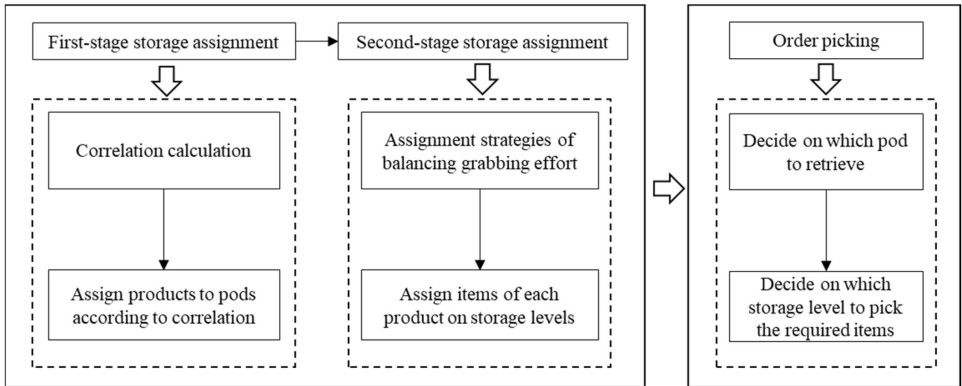

**Figure 4.** Solution framework of storage assignment and order picking.

*4.2. The First Stage of Optimization of Storage Assignment Based on FP-Growth Algorithm*

To improve retrieval efficiency, products are assigned to pods according to their descending-order correlation in the first stage of optimization. Highly correlated products are more likely to be assigned to the same pod. We adopt the FP-Growth algorithm to obtain the correlation between pairs of products from a large number of historic orders over a certain period. The calculation of $C_{j,j'}$ depends on the support degree. The support degree can be calculated for a single product but also between pairs of products. It indicates the frequency that a single product and pairs of products are requested in an order among all historical orders. The support degree is formulated in Equation (17), in which $F_j$ represents the number of orders that contain product $j \in J$, and $card(O)$ represents total number of orders. The correlation between products is accordingly obtained in Equation (18). In Equation (18), $Support\{j, j'\}$ is divided by $Support\{j\}$ and $Support\{j'\}$, because $Support\{j, j'\}$ could have a high value due to the high frequency of $Support\{j\}$. As such, the high value of $C_{j,j'}$ represents that product $j$ and product $j'$ are frequently ordered together.

$$Support\ \{j\} = \frac{F_j}{card(O)} \tag{17}$$

$$C_{j,j'} = \frac{Support\{j, j'\}}{Support\{j\} \times Support\{j'\}} \tag{18}$$

The products that are always required in a single order are described as single-ordered products. We filter the single-ordered products in historic orders and subsequently use the FP-Growth algorithm to calculate the frequency of correlated products. The thresholds of *Support* $\{j\}$ and $C_{j,j'}$ are set for filtering the products with the low correlation. The procedures of the FP-Growth algorithm include constructing the FP tree and mining frequent patterns from the FP tree. The steps of the FP-Growth algorithm are illustrated in Figure 5 and explained as follows:

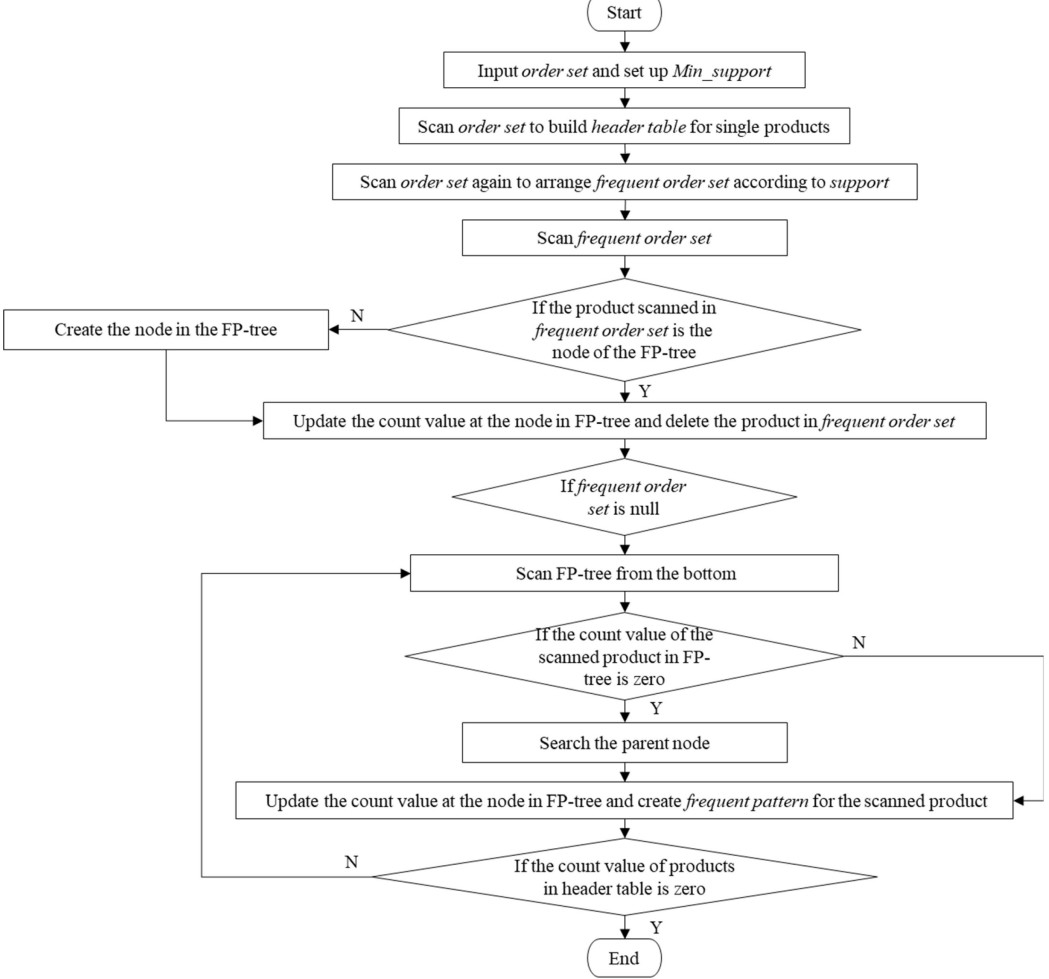

**Figure 5.** Steps of FP-Growth algorithm.

Step 1: Build the header table of products. Scan the order set $(O)$ once to count $F_j$, calculate *Support* $\{j\}$, set the threshold *Min_Support* for *Support* $\{j\}$, and filter product $j \in J$ if *Support* $\{j\}$ < *Min_Support*. Arrange product $j \in J$ according to *Support* $\{j\}$ in descending order and include the arranged product $j \in J$ and *Support* $\{j\}$ in the header table.

Step 2: Create a frequent order set. Scan the order set $(O)$ for the second time; delete products which are filtered in Step 1 from the order set $(O)$ and arrange the remaining products in descending order of *Support* $\{j\}$ to create the frequent order set.

Step 3: Build the FP tree. (1) Starting from creating the root node of the FP tree, scan the frequent order set. If the first product of the frequent order set is not the node of the FP tree, create a new root node for this product in the FP tree. (2) Otherwise, update the count value of this product at the existing node of the FP tree. (3) Repeat sub-steps of (1) and (2) until the occurrence of all products from the frequent order set are counted in the FP tree.

Step 4: Search for frequent patterns in the FP tree. (1) Start by targeting the bottom element of the header table. (2) Search the parent nodes of the targeted element in the FP tree. (3) The

route from the root node to the node of the targeted element in the FP tree is defined as a frequent pattern. As the correlation is calculated between two products, the number of products included into the frequent pattern is set as two. (4) Repeat sub-steps (1)–(3) until all products' frequent patterns are found in the FP tree.

Step 5: Obtain $Support\{j, j'\}$ based on the value of frequent patterns in the FP tree. (1) Search the value of the targeted element (product $j \in J$) in the header table. (2) Product $j' \in J$ is from the frequent pattern and $Support\{j, j'\}$ equals the value of product $j \in J$ searched from the header table.

$C_{j,j'}$ and $C_{j,p}$ are subsequently calculated according to the results obtained from the FP-Growth algorithm. To decide in which pod to store product $j \in J$ (the decision variable $x_{j,p}$), $C_{j,p}$ is calculated between product $j \in J$ to be assigned and products that have been assigned to a pod. The pod with the highest value of $C_{j,p}$ has the highest priority to store product $j \in J$. The procedure of assigning products to pods in the first stage of optimization is as shown in Figure 6.

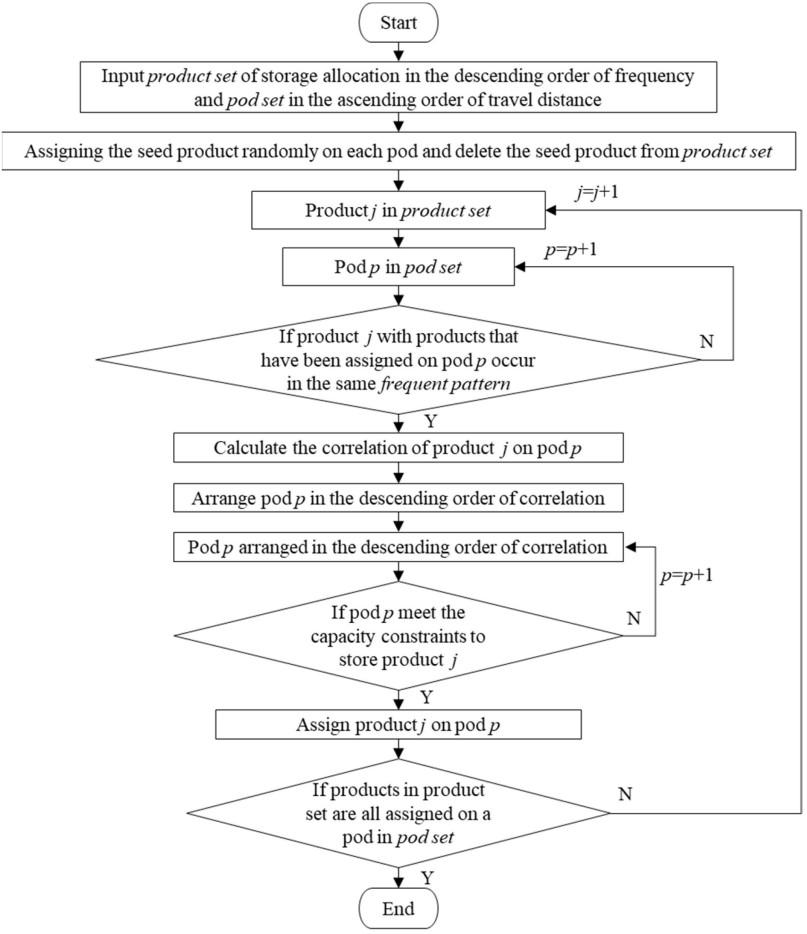

**Figure 6.** The procedure of assigning products to pods in the first stage of optimization.

### 4.3. The Second Stage of Optimization of Storage Assignment Based on Strategies of Balancing Grabbing Effort

The decision of assigning products to pods made in the first stage of optimization is the input of the second stage of optimization. We subsequently propose strategies for balancing the grabbing effort when assigning items of each product to storage levels in the second stage of optimization. The strategies of balancing the grabbing effort involve balancing the grabbing effort according to $(\alpha \cdot w_i + \beta \cdot v_i + \gamma \cdot l) \cdot t_{base}$. For example, assigning big and heavy items to the middle storage level can be part of the balanced grabbing effort alongside assigning the small and light items to high storage levels, with the assumption

that the grabbing effort at the high storage level is more than that of the middle storage level, and that the grabbing effort of big and heavy items is more than that of small and light items. We propose a weight-sorting strategy, a volume-sorting strategy, and a weight and volume-sorting strategy for balancing the grabbing effort in the second-stage storage assignment. In the process of assignment, the storage levels are prioritized in ascending order of $\gamma \cdot l$. For example, storage levels with a low value of $\gamma \cdot l$ have high priority in the storage assignment. The weight-sorting strategy involves arranging items of products in descending order of $\alpha \cdot w_i$; the volume-sorting strategy involves arranging items of products in descending order of $\beta \cdot v_i$; and the weight and volume-sorting strategy involves arranging items of products in descending order of $\alpha \cdot w_i + \beta \cdot v_i$. The arranged items are subsequently assigned to the storage levels, which are arranged by priority from high to low.

To evaluate the weight-sorting strategy, volume-sorting strategy, and weight- and volume-sorting strategy, we select the benchmark strategies which are commonly used in comparison in practice. The frequency-sorting strategy involves arranging items according to their frequency from high to low, so that human pickers spend less grabbing effort on picking frequent products. The stock-sorting strategy involves arranging items according to the total number items of the same products to be assigned. The largest number of items can be assigned to the storage level with the highest priority, in which the capacity of the prioritized storage level is fully used. The random strategy assigns items of each product randomly to storage levels for the easy operation of storage assignment in practice.

To explain the procedures of the weight-sorting strategy, volume-sorting strategy, and weight- and volume-sorting strategy, we take the instances of the low storage level, middle storage level, and high storage level. The assignment priority is middle storage level > low storage level > high storage level. The procedures of assigning items of each product are illustrated in Figure 7 and are explained as follows:

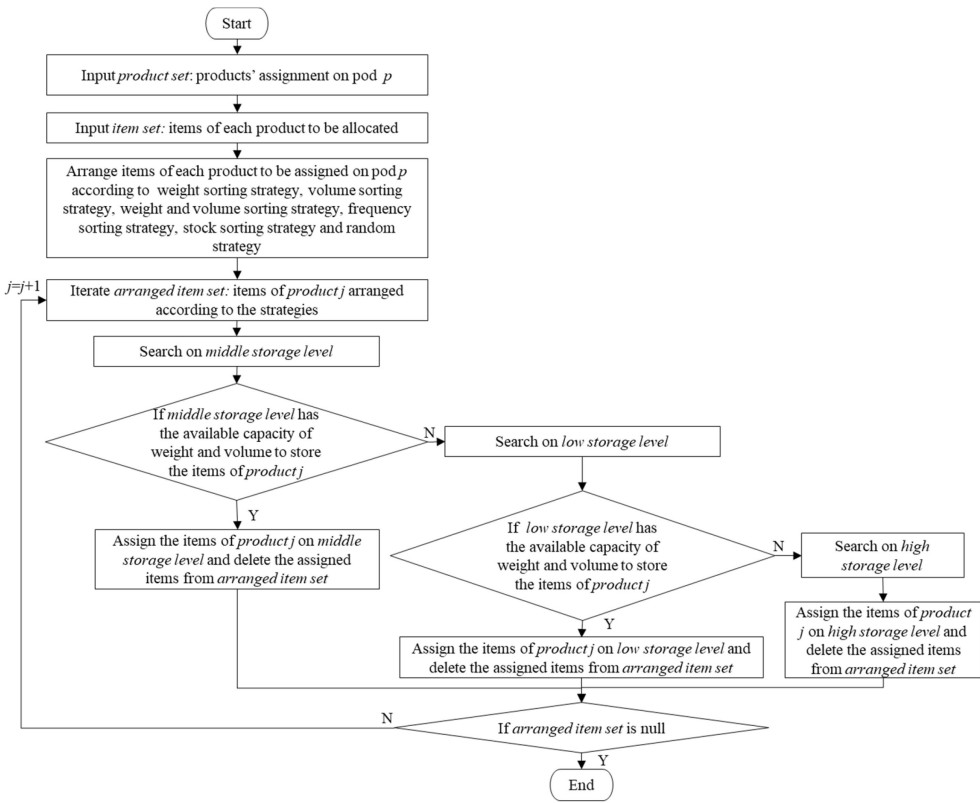

**Figure 7.** The procedures of assigning items of each product.

Step 1: Arrange item $i \in I_j$ of product $j \in J$ ($x_{j,p} = 1$) in descending order of the following indicators per strategy: $\alpha \cdot w_i$ in the weight-sorting strategy; $\beta \cdot v_i$ in the volume-sorting strategy;

$\alpha \cdot w_i + \beta \cdot v_i$ in the weight- and volume-sorting strategy; $F_j$ in the frequency-sorting strategy; $s_j$ in the stock-sorting strategy; and a random arrangement in the random strategy.

Step 2: Prioritize storage level $l \in L$. The middle storage level, which is at a height similar to human pickers and easiest to access, has the highest priority. The low storage level, which requires bending to reach, is given less priority than the middle storage level, but has more priority than the high storage level from which human pickers need to climb a ladder to pick.

Step 3: Assign the arranged items of each product according to the priority and the capacity constraints of each storage level. For example, if the middle storage level has the available capacity in terms of weight and volume to store the items to be assigned, the decision of assigning the items of the product can be made consequently. Otherwise, the low storage level and high storage level are searched to make the assignment decision on which has the available capacity in terms of weight and volume to store the items.

## 5. Numerical Analysis

This section analyzes the solutions of storage assignment obtained from the two-stage optimization. We first introduce a real-order dataset and a sketched layout of the warehouse used in the numerical analysis. The solution derived from the correlated storage assignment is compared with class-based storage assignment. The solutions obtained through the strategies of balancing the grabbing effort are evaluated based on the retrieval time and grabbing time. The effectiveness of the two-stage optimization is shown in an analysis compared to random assignment. A sensitivity analysis is conducted regarding the influence coefficients $\alpha$, $\beta$, and $\gamma$ on the grabbing time. We also derive some management insights in the discussion.

### 5.1. Dataset and Layout Description

We use 200,000 historical orders to calculate correlations between products. All historical orders are from real customer purchases of an e-commerce company. This company operates its own warehouse in order to quickly respond to customers' demands. The layout of the company's warehouse is sketched in Figure 8. Pods are located in the storage area and workstations are located at the two sides of the storage area. Robots drive the pods at a constant speed of $v = 2\,m/s$. The pod is served by the closest workstation and $d_p$ is calculated by the Manhattan distance. In total, 1200 pods are available for storage assignment in the empty warehouse and are the same size. The parameters set for each pod are $M = 40$, $N = 400$, $W = 100kg$, and $V = 2\,\text{m}^3$. To decrease the complexity of solving the instance, we set three storage levels for each pod ($L = \{1, 2, 3\}$), in which we denote 1 as the middle storage level, 2 as the low storage level, and 3 as the high storage level according to the priority rules of storage assignment. Based on human pickers' experience working in the warehouse, grabbing an item takes at least one second. We therefore set $t_{base} = 1$ s. The influence coefficients of the weight, volume, and storage level of the pod are set as $\alpha = 1, \beta = 1$, and $\gamma = 1$ which are further discussed in the sensitivity analysis. To filter out single-ordered products from the correlated products in the order set, $Min\_Support = 3$ is input into the FP-Growth algorithm. As a result, 11,000 products are highly correlated with each other and 30,000 products are of less frequency than $Min\_Support$. Hence, 11,000 products are used in correlated storage assignment and the other 30,000 products are assigned randomly.

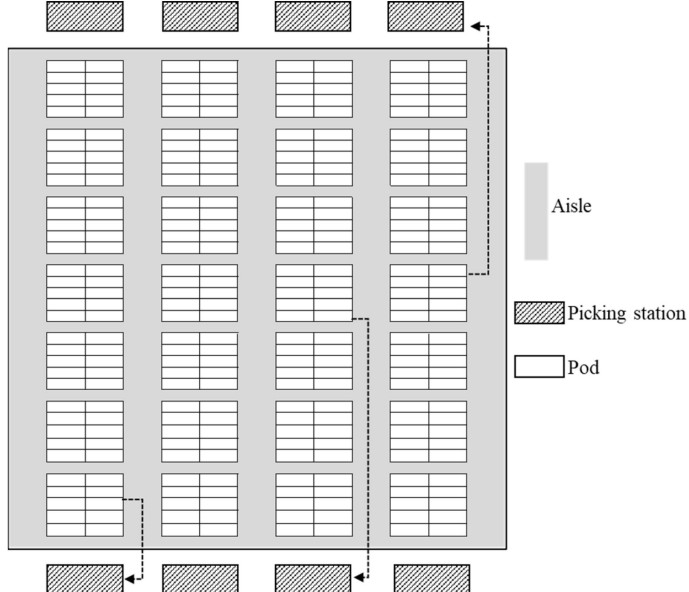

**Figure 8.** The sketched layout of the warehouse.

*5.2. Results and Discussion*

In this section, we first analyze the order-picking efficiency under the first stage of optimization of the storage assignment, in which correlated storage assignment and class-based storage assignment are compared. We subsequently compare the solutions obtained under the two-stage storage assignment optimization. Solutions under the strategies of balancing the grabbing effort are compared with benchmarks, in which the effectiveness is shown in the comparison to random assignment. The grabbing time for each storage level is specifically analyzed under the strategies of balancing the grabbing effort and benchmarks. We additionally compare the capacity usage of weight and volume on each storage level and analyze the advantages and disadvantages of prioritizing storage levels in the storage assignment. In the sensitivity analysis, the order-picking efficiencies are compared under influence coefficients $\alpha$, $\beta$, and $\gamma$.

5.2.1. The Compared Results in the First Stage of Optimization of Storage Assignment

To analyze the results of the first stage of storage assignment optimization, we compare the total order-picking time under correlated storage assignment and class-based storage assignment. The total order-picking time is calculated by adding the retrieval time and the grabbing time. In the analysis of the first stage of storage assignment optimization, the assignment of products to pods is optimized and the items of each product are randomly assigned to the storage levels. Therefore, the grabbing time is the same under correlated storage assignment and class-based storage assignment, but the retrieval time makes a difference.

Class-based storage assignment is commonly used in practice. In class-based storage assignment, products are classified based on their turnover rate and assigned to the storage area of the corresponding class. Subsequently, products are randomly assigned to the specific storage locations in the storage area. ABC classification is the most common, in which fast-moving products are stored in class A; slow-moving products are stored in class B; and long-tail products are stored in class C. The total order-picking time under correlated storage assignment and class-based storage assignment is compared in Figure 9. The number of pod retrievals under correlated storage assignment and class-based storage assignment is compared in Figure 10. Correlated storage assignment effectively reduces the total order-picking time compared to class-based storage assignment. To further analyze, a decrease in pod retrievals leads to the improvement of order-picking efficiency in correlated storage assignment, since correlated products which are stored together in the same pod can also be picked together in one retrieval. This result also shows that correlated storage

assignment plays an important role in improving order-picking efficiency. The large decrease in pod retrievals under correlated storage assignment indicates that orders can be picked with less energy consumption of the mobile robots, which is beneficial for sustainable warehousing management in human–robot collaboration.

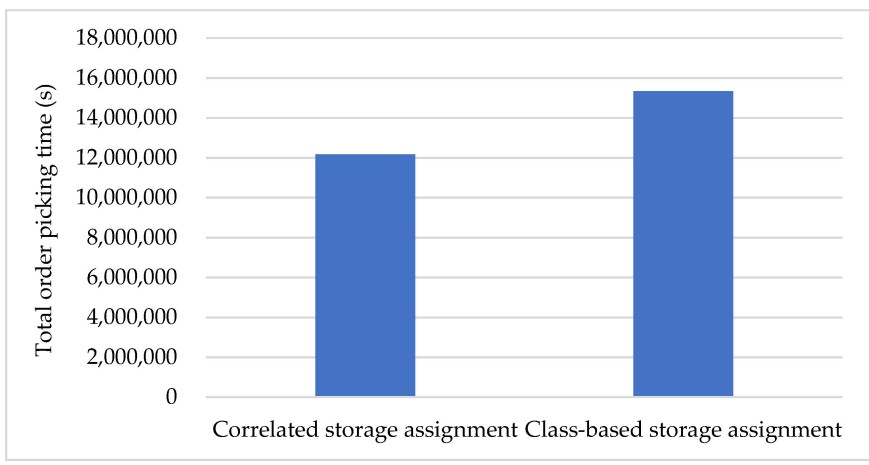

**Figure 9.** Total order-picking time in correlated storage assignment and class-based storage assignment.

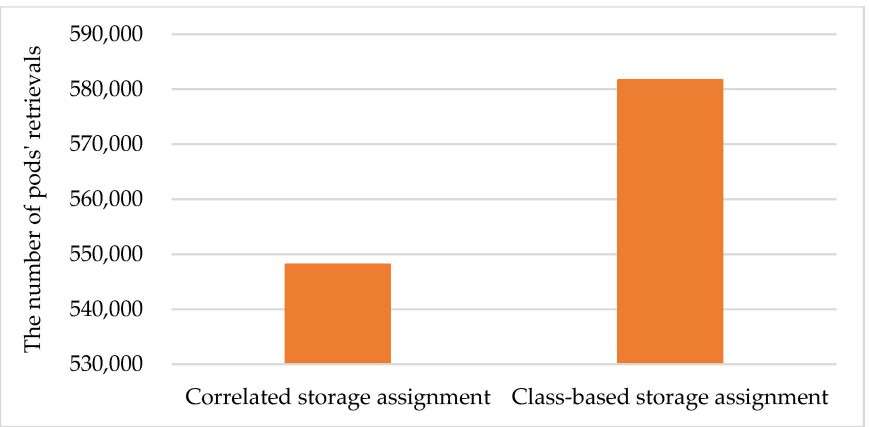

**Figure 10.** The number of pod retrievals in correlated storage assignment and class-based storage assignment.

### 5.2.2. The Effectiveness of the Two-Stage Optimization of Storage Assignment

Based on the assignment of products obtained under correlated storage assignment, the total order-picking time is compared under the various strategies of assigning items to storage levels. The strategies of weight sorting, volume sorting, weight and volume sorting, frequency sorting, and stock sorting are used in the two-stage storage assignment optimization. Random assignment is regarded as only being optimized in the first stage. Compared to the total order-picking time under random assignment, the effectiveness of the two-stage storage assignment optimization is shown in Figure 11. The total order-picking time is highly reduced under the strategies used in the two-stage optimization of storage assignment compared to random assignment. In other words, the two-stage optimization of storage assignment effectively improves order-picking efficiency. Compared with the total order-picking time under the benchmark strategies of frequency sorting and stock sorting, the advantages of the weight-sorting and volume-sorting strategies are not obvious. It provides the management insight that the frequency-sorting and stock-sorting strategies are of similar effectiveness to the weight-sorting and volume-sorting strategies when balancing the grabbing effort among the storage levels. The weight- and volume-sorting strategy performs the best among all strategies. We can conclude that taking items' weight and

volume into account when assigning items of each product to storage levels improves the order-picking efficiency.

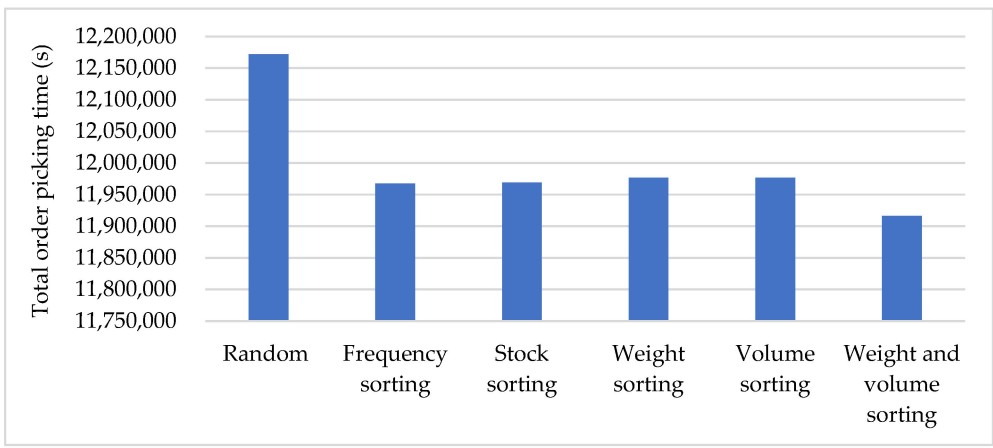

**Figure 11.** Total order-picking time under the storage assignment strategies.

To further analyze how grabbing efforts spent by a human picker are distributed among the middle storage level, low storage level, and high storage level, the grabbing time for each storage level is depicted in Figure 12 by storage assignment strategy. The grabbing time is almost evenly distributed on the middle storage level, low storage level, and high storage level under random assignment. The reason is that the storage levels are not prioritized in the random assignment of items. In the results obtained under the strategies used in the two-stage optimization, the grabbing time spent on the high storage level is much less than that on the middle storage level and low storage level. Hence, we can obtain the management insight that less grabbing time spent on the high storage level benefits the order-picking efficiency more. Compared to random assignment, the shorter grabbing time under the weight and volume storage assignment strategy indicates that human pickers spend less grabbing effort to pick the required items, which leads to the sustainable performance of human pickers in human–robot collaboration.

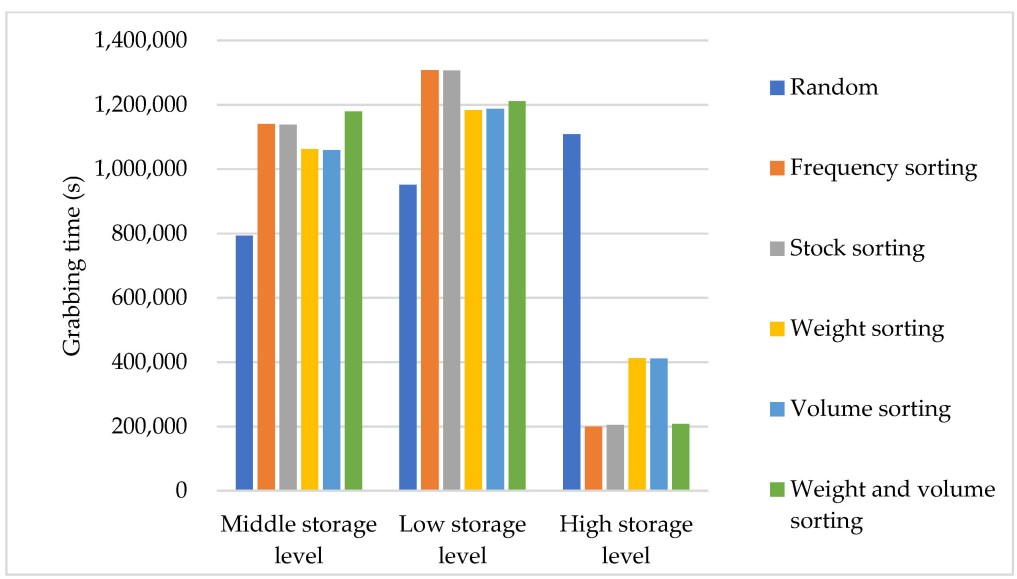

**Figure 12.** Grabbing time on each storage level under the storage assignment strategies.

### 5.2.3. Analysis of Capacity Usage on the Storage Levels

The middle storage level and low storage level are highly prioritized in the strategies of balancing the grabbing effort. On the other hand, this also leads to an imbalanced capacity usage of each storage level, as shown in Table 2. We analyze the capacity usage on each storage level regarding the aspects of weight and volume. Random storage assignment has a balanced capacity usage of weight (66%, 66%, and 65.7%) and volume (65.9%, 65.5%, and 65.7%) on the middle storage level, low storage level, and high storage level. Although random storage assignment weakens the order-picking efficiency, the balanced capacity usage could save the capacity of a pod and the floor space of the storage area. As a result, the size of pods can be reduced by 35%, which is almost equal to the percentage of the remaining capacity on each storage level.

**Table 2.** Capacity usage of weight and volume on each storage level.

| | Capacity Usage of Weight | | | | | |
|---|---|---|---|---|---|---|
| | Random | Frequency Sorting | Stock Sorting | Weight Sorting | Volume Sorting | Weight and Volume Sorting |
| Middle storage level | 66.0% | 94.8% | 94.6% | 70.4% | 99.4% | 94.1% |
| Low storage level | 66.0% | 90.6% | 90.6% | 95.1% | 77.0% | 89.3% |
| High storage level | 65.7% | 12.3% | 12.5% | 32.3% | 21.4% | 14.3% |
| | Capacity usage of volume | | | | | |
| | Random | Frequency Sorting | Stock Sorting | Weight Sorting | Volume Sorting | Weight and Volume Sorting |
| Middle storage level | 65.9% | 94.3% | 94.2% | 99.3% | 69.7% | 93.2% |
| Low storage level | 65.5% | 90.8% | 90.6% | 76.6% | 95.2% | 89.7% |
| High storage level | 65.7% | 12.0% | 12.4% | 21.3% | 32.1% | 14.3% |

The capacity usage of weight and volume on the middle storage level and low storage level dominates the capacity usage of a pod in the strategies of frequency sorting, stock sorting, and weight and volume sorting. The capacity usage of weight and volume on the middle storage level and low storage level dominates the capacity of a pod, but only a few capacities are used on the high storage level, in the strategies of frequency sorting, stock sorting, and weight and volume sorting. We therefore suggest that the high storage level can be used for storing products rather than order picking in these strategies. The imbalanced capacity usage of weight and volume on each storage level is also apparent in the weight-sorting strategy and volume-sorting strategy. We conclude that random storage assignment takes the most advantage of the capacity usage of weight and volume; the strategies of frequency sorting, stock sorting, and weight and volume sorting can also effectively manage the capacity usage of a pod when the high storage level is used for storing and replenishment; and the weight-sorting strategy and volume-sorting strategy are disadvantaged in terms of capacity management. Saving the capacity usage under random storage assignment also brings the benefit of sustainable resource management in warehouses.

### 5.2.4. Sensitivity Analysis

The influence coefficients are set as $\alpha = 1, \beta = 1$, and $\gamma = 1$ in the initial solution. To analyze which of the influence coefficients has the greatest effect on the total order-picking time, we set the three groups of influence coefficients as $\alpha = 1, \beta = 1, \gamma = 0.5; \alpha = 1,$ $\beta = 0.5, \gamma = 1$; and $\alpha = 0.5, \beta = 1, \gamma = 1$, in which one parameter of $\alpha, \beta, \gamma$ is changed in one group. Table 3 shows the sensitivity analysis of $\alpha, \beta, \gamma$ on the total order-picking time. Compared with the initial setting, $\alpha = 1, \beta = 1, \gamma = 1$, the improvement of the total

order-picking time at $\alpha = 1$, $\beta = 1$, and $\gamma = 0.5$ is similar to that in $\alpha = 1$, $\beta = 0.5$, and $\gamma = 1$. This percentage is also almost the same at $\alpha = 0.5$, $\beta = 1$, and $\gamma = 1$. This implies that the influence coefficients $\alpha$, $\beta$, and $\gamma$ have similar effects on the total order-picking time, and this trend is not affected by the storage assignment strategies adopted in this study.

**Table 3.** Sensitivity analysis of influence coefficients of $\alpha$, $\beta$, and $\gamma$ on total time.

| | **Total Time (s)** | | | | | |
|---|---|---|---|---|---|---|
| **$\alpha,\beta,\gamma$** | **Random** | **Frequency Sorting** | **Stock Sorting** | **Weight Sorting** | **Volume Sorting** | **Weight and Volume Sorting** |
| $\alpha = 1, \beta = 1, \gamma = 1$ | 12,171,875.5 | 11,967,660.5 | 11,969,347.5 | 11,976,691.5 | 11,976,657.5 | 11,916,219.5 |
| $\alpha = 1, \beta = 1, \gamma = 0.5$ | 11,693,167 | 11,591,059.5 | 11,591,903 | 11,595,575 | 11,595,558 | 11,565,339 |
| Improvement (%) | (−3.9%) | (−3.1%) | (−3.2%) | (−3.2%) | (−3.2%) | (−2.9%) |
| $\alpha = 1, \beta = 0.5, \gamma = 1$ | 11,698,730.5 | 11,494,515.5 | 11,496,202.5 | 11,503,546.5 | 11,503,512.5 | 11,443,074.5 |
| Improvement (%) | (−3.9%) | (−4.0%) | (−4.0%) | (−4.0%) | (−4.0%) | (−4.0%) |
| $\alpha = 0.5, \beta = 1, \gamma = 1$ | 11,697,280 | 11,493,065 | 11,494,752 | 11,502,096 | 11,502,062 | 11,441,624 |
| Improvement (%) | (−3.9%) | (−4.0%) | (−4.0%) | (−4.0%) | (−4.0%) | (−4.0%) |

To further analyze how the influence coefficients affect the grabbing time on each storage level, we compare the variation in the grabbing time specifically on each storage level, as shown in Table 4. In the compared results of the middle storage level, $\alpha = 0.5$, $\beta = 1$, and $\gamma = 1$ have the greatest effect on the grabbing time in the strategies of random assignment, frequency-sorting, stock-sorting, volume-sorting, and weight- and volume-sorting, whereas $\alpha = 1$, $\beta = 0.5$, and $\gamma = 1$ bring the most improvement in the weight-sorting strategy. This result shows that an item's weight has the greatest effect on the order-picking efficiency of the middle storage level, but the exception for this is the weight-sorting strategy. This result, on the other hand, indicates that the decrease in $\beta$ weakens the advantage of the weight-sorting strategy on the order-picking efficiency of the middle storage level.

**Table 4.** Sensitivity analysis of influence coefficients of $\alpha$, $\beta$, and $\gamma$ on grabbing time.

| | Grabbing Time on Middle Storage Level (s) | | | | | |
|---|---|---|---|---|---|---|
| $\alpha, \beta, \gamma$ | Random | Frequency Sorting | Stock Sorting | Weight Sorting | Volume Sorting | Weight and Volume Sorting |
| $\alpha = 1, \beta = 1, \gamma = 1$ | 793,358 | 1,140,670 | 1,138,027 | 1,061,895 | 1,059,226 | 1,179,067 |
| $\alpha = 1, \beta = 1, \gamma = 0.5$ | 713,334.5 | 1,024,286 | 1,022,032 | 937,998 | 935,465 | 1,039,108 |
| Improvement (%) | (−10.1%) | (−10.2%) | (−10.2%) | (−11.7%) | (−11.7%) | (−11.9%) |
| $\alpha = 1, \beta = 0.5, \gamma = 1$ | 635,142.5 | 914,349 | 912,035.5 | 823,694.5 | 891,826.5 | 955,399.5 |
| Improvement (%) | (−19.9%) | (−19.8%) | (−19.9%) | (−22.4%) | (−15.8%) | (−19.0%) |
| $\alpha = 0.5, \beta = 1, \gamma = 1$ | 634,918 | 913,040 | 911,000.5 | 893,045 | 820,773.5 | 953,160 |
| Improvement (%) | (−20.0%) | (−20.0%) | (−19.9%) | (−15.9%) | (−22.5%) | (−19.2%) |
| | Grabbing time on low storage level (s) | | | | | |
| $\alpha, \beta, \gamma$ | Random | Frequency Sorting | Stock Sorting | Weight Sorting | Volume Sorting | Weight and Volume Sorting |
| $\alpha = 1, \beta = 1, \gamma = 1$ | 951,204 | 130,8073 | 1,307,103 | 1,183,214 | 1,187,004 | 1,210,654 |
| $\alpha = 1, \beta = 1, \gamma = 0.5$ | 791,310 | 1,089,406 | 1,088,569 | 1,003,630 | 1,006,842 | 1,034,846 |
| Improvement (%) | (−16.8%) | (−16.7%) | (−16.7%) | (−15.2%) | (−15.2%) | (−14.5%) |
| $\alpha = 1, \beta = 0.5, \gamma = 1$ | 793,890.5 | 1,090,134 | 1,089,645 | 999,411 | 958,405 | 995,425 |
| Improvement (%) | (−16.5%) | (−16.7%) | (−16.6%) | (−15.5%) | (−19.3%) | (−17.8%) |
| $\alpha = 0.5, \beta = 1, \gamma = 1$ | 792,809.5 | 1,090,643 | 1,089,544 | 954,994 | 1,002,263 | 996,364 |
| Improvement (%) | (−16.7%) | (−16.6%) | (−16.6%) | (−19.3%) | (−15.6%) | (−17.7%) |

**Table 4.** *Cont.*

| | Grabbing time on high storage level (s) | | | | | |
|---|---|---|---|---|---|---|
| | Random | Frequency Sorting | Stock Sorting | Weight Sorting | Volume Sorting | Weight and Volume Sorting |
| $\alpha = 1, \beta = 1, \gamma = 1$ | 110,8336 | 199,940 | 205,240 | 412,605 | 411,450 | 207,521 |
| $\alpha = 1, \beta = 1, \gamma = 0.5$ | 869,545 | 158,390 | 162,325 | 334,969.5 | 334,273.5 | 172,407.5 |
| Improvement (%) | (−21.5%) | (−20.8%) | (−20.9%) | (−18.8%) | (−18.8%) | (−16.9%) |
| $\alpha = 1, \beta = 0.5, \gamma = 1$ | 950,720 | 171,055 | 175,545 | 361,463.5 | 334,303.5 | 173,272.5 |
| Improvement (%) | (−14.2%) | (−14.4%) | (−14.5%) | (−12.4%) | (−18.7%) | (−16.5%) |
| $\alpha = 0.5, \beta = 1, \gamma = 1$ | 950,575 | 170,405 | 175,230 | 335,079.5 | 360,048 | 173,122.5 |
| Improvement (%) | (−14.2%) | (−14.8%) | (−14.6%) | (−18.8%) | (−12.5%) | (−16.6%) |

Comparing the results of the grabbing time on the low storage level, the improvements in the grabbing time at $\alpha = 1, \beta = 1,$ and $\gamma = 0.5$; $\alpha = 1, \beta = 0.5,$ and $\gamma = 1$; and $\alpha = 0.5, \beta = 1,$ and $\gamma = 1$, are almost the same in the strategies of random assignment, frequency sorting, and stock sorting. This implies that $\alpha, \beta,$ and $\gamma$ have equal importance in the order-picking efficiency of the low storage level in the strategies of random assignment, frequency sorting, and stock sorting. Regarding the grabbing time on the low storage level, the greatest improvement appears at $\alpha = 0.5, \beta = 1,$ and $\gamma = 1$ in the weight-sorting strategy; at $\alpha = 1, \beta = 0.5,$ and $\gamma = 1$ in the volume-sorting strategy; and at both $\alpha = 0.5, \beta = 1,$ and $\gamma = 1$ and $\alpha = 1, \beta = 0.5,$ and $\gamma = 1$ in the weight- and volume-sorting strategy. The implication can be obtained that the decrease in $\beta$ and $\gamma$ strengthens the advantages of the weight-sorting strategy, volume-sorting strategy, and weight- and volume-sorting strategy on the order-picking efficiency of the low storage level. This observation is opposite to the sensitive results of the grabbing time on the middle storage level.

The greatest improvement in the grabbing time on the high storage level is obtained at $\alpha = 1, \beta = 1,$ and $\gamma = 0.5$ in all strategies of storage assignment. This suggests that $\gamma$ has more importance than $\alpha$ and $\beta$ in the order-picking efficiency of the high storage level.

**6. Conclusions, Limitations, and Future Research**

This study defines the storage assignment allocation (SLA) problem as assigning items of products to the slots of pods in an RMF system. The SLA problem is solved by the two-stage optimization method for the purpose of improving the order-picking efficiency in RMF systems. Most research only considers the retrieval of pods in the order-picking process. Given that human pickers play an important role in human–robot collaborative order picking, we consider both the retrieval of pods and human pickers' operations to evaluate order-picking efficiency. Additionally, we consider the grabbing behavior of human pickers to measure the operation efficiency of human pickers. The storage levels on which the slots are placed and a product's weight and volume are the main factors that influence the grabbing time of human pickers. We therefore included these factors to measure human pickers' grabbing time. Due to the NP-hardness of the SLA problem, we proposed a two-stage optimization method and designed heuristic algorithms based on strategies of balancing the grabbing effort. The decision of assigning products to pods is made in the first stage of optimization according to correlated storage assignment. Assigning items of each product to storage levels is subsequently decided, in which the weight-sorting strategy, volume-sorting strategy, and weight- and volume-sorting strategy are adopted.

Our findings show that correlated storage assignment more greatly benefits the retrieval efficiency of pods than class-based storage assignment. And including behavioral factors in storage location assignment improves the operation efficiency of human pickers in RMF systems compared to random storage assignment. The retrieval efficiency of pods and the operation efficiency of human pickers are both improved, which benefits sustainable warehousing operations in the long run. More specifically, the improvement of order-picking efficiency results from the decrease in pod retrievals and the shorter grabbing time spent by human pickers. This leads to the sustainable energy consumption of

mobile robots and the sustainable performance of human pickers, which further benefits sustainable human–robot collaboration.

The main limitation of this study is that human pickers pick at fixed workstations and the pods are retrieved to the fixed and closest workstation. The workload balanced among workstations also affects human–robot collaborative order-picking efficiency in RMF systems. The assignment of pods to workstations can be jointly optimized with the storage assignment with a balanced workload in future research. Another limitation is that the FP-Growth algorithm, adopted in this study, is only used to calculate the correlated relationship between pairs of products. Correlated relationships may exist among multiple products in an order. Adaptations to the FP-Growth algorithm are essential for future research.

**Author Contributions:** Conceptualization, Y.C. and Y.L.; methodology, Y.C. and Y.L.; investigation, Y.C. and Y.L.; resources, Y.C. and Y.L.; data curation, Y.C. and Y.L.; writing—original draft preparation, Y.C. and Y.L.; writing—review and editing, Y.C. and Y.L.; visualization, Y.C. and Y.L.; supervision, Y.L.; project administration, Y.L.; funding acquisition, Y.L. All authors have read and agreed to the published version of the manuscript.

**Funding:** This research was supported by the National Natural Science Foundation of China (NSFC) under grant number 71831001 and the Beijing Logistics Informatics Research Base.

**Institutional Review Board Statement:** Not applicable.

**Informed Consent Statement:** Not applicable.

**Data Availability Statement:** Data are contained within the article.

**Conflicts of Interest:** The authors declare no conflicts of interest.

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
