# Peer review of "Storage Location Assignment for Improving Human–Robot Collaborative Order-Picking Efficiency in Robotic Mobile Fulfillment Systems"

_sustainability, doi:10.3390/su16051742_

Round 1
Reviewer 1 Report
Comments and Suggestions for Authors
The article addresses current and crucial aspects related to warehouse efficiency, shedding light on contemporary challenges and potential solutions. The authors provide valuable insights into optimizing operations, leveraging cutting-edge technologies, cooperating people with robots, and tackling emerging trends. Overall, it is a well-researched and insightful piece that captures the essence of critical considerations for enhancing warehouse productivity.
While the article provides a comprehensive overview of essential aspects and research, a few points warrant further attention from the author. It is recommended that the authors consider the following aspects:
Add to Table 1 the set of items of jth type of product Ij. It is essential for the following formulations of the mathematical model, e.g. wi.
Use the N, M, W, and V parameters for the entire pod, and all types of products may be too general. Please consider making these parameters dependent on the product type (j)
If si is the total number of items of the jth product that should be placed in the pods and xj,p takes the value {0,1}, then the constraint (3) may not be valid. It forces all items of this product to be in only one pod. What if there are more items than the capacity of one pod?
There is no information about l in constraint (6). For each l?
Is assigning items of the same product to more than one level forbidden (constraint (9))? Why?
Improve the formatting of table headers (column width, line spacing, etc.)
Change the description in the charts (Fig. 9-10) from srorage to storage assignment.
Reviewer 2 Report
Comments and Suggestions for Authors
- What is the main question addressed by the research?
Although the problem was raised, the research questions that will be answered in the study were not presented. The problem needs to be clearly defined. Research questions, objectives, and contribution should be clearly defined in the introduction section.
- Do you consider the topic original or relevant in the field? Does it address a specific gap in the field?
Although the topic in general is not new, the paper proposes a different solution methodology with additional parameter. Therefore, research field and idea is considered related to the topic of the journal and presents a solution in another way. In order to increase the efficiency of human pickers' work in RMF systems and the pods' ability to retrieve goods, the article helps solve the SLA problem by assigning products to the pods' slots. A two-phase optimization approach is suggested to address the SLA problem, wherein heuristic algorithms employ correlated storage assignment and balanced grabbing effort tactics.
- What does it add to the subject area compared with other published aterial?
The article presents the process of assigning product items in order to solve SLA problems. Furthermore, decisions are made regarding which pod to store, and more precisely, which pod slot to store. The complexity of solving the SLA problem increases as a result of these decisions. To overcome the issue, the paper suggests a two-step optimization process. Products are assigned to pods in the first stage of storage assignment, when the heuristic method uses correlated storage assignment. In the second step of storage assignment, where solutions for balancing physical effort among storage levels are given for the heuristic algorithm design, it is chosen which items to allocate to each product on the slots.
- What specific improvements should the authors consider regarding the methodology? What further controls should be considered?
Although the method used is clear, sequential, and smooth, the applications are not clear, and I believe that there is difficulty in the practical application of this method. To further improve pod retrieval efficiency and human picker operating efficiency in RMF systems, the assignment of pods to workstations can be taken into consideration in conjunction with storage assignment and workload balancing. The main concern is how the authors validate the algorithm.
- Are the conclusions consistent with the evidence and arguments presented and do they address the main question posed?
The conclusions make sense in light of the facts and arguments put out, and they cover the main issue. However, authors also need to take into account the research limitations, applicability, and further future research discussion.
- Are the references appropriate?
The references used are appropriate and up-to-date and cover most parts of the research
7. Please include any additional comments on the tables and figures.
Tables and figures are appropriate and clear.
Reviewer 3 Report
Comments and Suggestions for Authors
The paper aims at solving problem of Storage Location Assignment (SLA) in Robot Mobile Fulfillment RMF systems with human pickers.
The topic is interesting and up to date. The contribution to sustainability research is limited.
The paper requires revisions as follows:
· Please rewrite the abstract to better highlight the background/motivation for this study, research gap, aim, and contributions of the study. Explain why your research is original and novel?
· In the Introduction section, highlight more the place of this study in a broader context and highlight why it is important. Describe more the originality of your approach. Link your research more with sustainability. Explain more clearly what is the aim is of this research.
· The methodology is not described sufficiently. Better justify your approach to two-stage optimization. What are the benefits and limitations of using the Fp-Growth Algorithm in first-stage optimization? Justify the choice of tools and methods for the second stage of optimization? It is not clear how your mathematical model supports decision making in sustainable warehousing.
· Discussion of the results is limited. Please explain how your results might be relevant in a broader context. How can they be used to make warehouse management more sustainable?
· In the conclusion you have written “This implies the effectiveness of two-stage optimization method in solving SLA problem defined in this paper and the improvement of order picking efficiency benefits sustainable warehousing operations in the long run” How?? It is not clear, make sure that the link with sustainable warehousing is properly explained.
· In the conclusions, highlight originality of the findings (in comparison with the state of the art), limitations, or you research
·
Comments on the Quality of English Language
The paper aims at solving problem of Storage Location Assignment (SLA) in Robot Mobile Fulfillment RMF systems with human pickers.
The topic is interesting and up to date. The contribution to sustainability research is limited.
The paper requires revisions as follows:
· Please rewrite the abstract to better highlight the background/motivation for this study, research gap, aim, and contributions of the study. Explain why your research is original and novel?
· In the Introduction section, highlight more the place of this study in a broader context and highlight why it is important. Describe more the originality of your approach. Link your research more with sustainability. Explain more clearly what is the aim is of this research.
· The methodology is not described sufficiently. Better justify your approach to two-stage optimization. What are the benefits and limitations of using the Fp-Growth Algorithm in first-stage optimization? Justify the choice of tools and methods for the second stage of optimization? It is not clear how your mathematical model supports decision making in sustainable warehousing.
· Discussion of the results is limited. Please explain how your results might be relevant in a broader context. How can they be used to make warehouse management more sustainable?
· In the conclusion you have written “This implies the effectiveness of two-stage optimization method in solving SLA problem defined in this paper and the improvement of order picking efficiency benefits sustainable warehousing operations in the long run” How?? It is not clear, make sure that the link with sustainable warehousing is properly explained.
· In the conclusions, highlight originality of the findings (in comparison with the state of the art), limitations, or you research
· Please proofread the paper to improve readability
Round 2
Reviewer 3 Report
Comments and Suggestions for Authors
I appreciate the time and effort the authors have put to improve the article and to reply to my comments.
The revised version of the article is improved and more suitable for publication. However, the link with sustainability research is still rather weak. I recommend that the authors extend the explanation about how their research contributes to sustainable warehousing.
Comments on the Quality of English LanguageEnglish shall be improved.
Author Response
We thank the reviewer for the effort and time spent on reviewing the revised manuscript.
We explain the contribution of this research to sustainable warehousing from the following points:
(1) RMF system is the human-robot collaborative order picking system. Compared to manual order picking, RMF system largely decreases human pickers’ physical effort and improves order picking efficiency. Human pickers could spend less working time, which leading to cutting the carbon footprint.
(2) The purpose of this research is to improve the human-robot collaborative order picking efficiency (measured by travel time) in RMF system. The decrease of travel time in RMF system also represents picking the same number of orders with less energy consumption. The corresponds to the call for sustainable energy consumption in warehouse.
(3) Human behavioral factors, which are also defined as ergonomical factors (e.g. bending or stretching to retrieve items from locations) are involved in the storage assignment optimization. Taking human picker’s work-related well-being into account also aligns with the goal of sustainable development goals in warehouse.